# First Proteomic Approach to Identify Cell Death Biomarkers in Wine Yeasts during Sparkling Wine Production

**DOI:** 10.3390/microorganisms7110542

**Published:** 2019-11-08

**Authors:** Juan Antonio Porras-Agüera, Jaime Moreno-García, Juan Carlos Mauricio, Juan Moreno, Teresa García-Martínez

**Affiliations:** 1Department of Microbiology, Agrifood Campus of International Excellence ceiA3, University of Cordoba, 14014 Cordoba, Spain; b02poagj@uco.es (J.A.P.-A.); b62mogaj@uco.es (J.M.-G.); mi2gamam@uco.es (T.G.-M.); 2Department of Agricultural Chemistry, Agrifood Campus of International Excellence ceiA3, University of Cordoba, 14014 Cordoba, Spain; qe1movij@uco.es

**Keywords:** apoptosis, autolysis, proteome, yeast, sparkling wine, CO_2_ overpressure

## Abstract

Apoptosis and later autolysis are biological processes which take place in *Saccharomyces cerevisiae* during industrial fermentation processes, which involve costly and time-consuming aging periods. Therefore, the identification of potential cell death biomarkers can contribute to the creation of a long-term strategy in order to improve and accelerate the winemaking process. Here, we performed a proteomic analysis based on the detection of possible apoptosis and autolysis protein biomarkers in two industrial yeast strains commonly used in post-fermentative processes (sparkling wine secondary fermentation and biological aging) under typical sparkling wine elaboration conditions. Pressure had a negatively effect on viability for flor yeast, whereas the sparkling wine strain seems to be more adapted to these conditions. Flor yeast strain experienced an increase in content of apoptosis-related proteins, glucanases and vacuolar proteases at the first month of aging. Significant correlations between viability and apoptosis proteins were established in both yeast strains. Multivariate analysis based on the proteome of each process allowed to distinguish among samples and strains. The proteomic profile obtained in this study could provide useful information on the selection of wine strains and yeast behavior during sparkling wine elaboration. Additionally, the use of flor yeasts for sparkling wine improvement and elaboration is proposed.

## 1. Introduction

Sparkling wines’ production involves a secondary fermentation of a base wine followed by a prolonged aging period in contact with lees. Along this stage, known as “prise de mousse”, yeast cells must endure hard stress conditions such as high ethanol concentration, nutrient starvation, low pH and temperature, and CO_2_ overpressure (6 bars) [1]. Such stress conditions compromise and test the yeast survival, which is ideal to analyze cell death-related processes. Among different yeasts, *Saccharomyces cerevisiae*, due to its easy handling and tractability, has been widely used to study cell death [2,3,4].

Programmed cell death (PCD) represents a specific type of regulated cell death (RCD), which occurs in strictly physiological scenarios such as development or aging. PCD has been reported to occur with apoptotic features [3,5]. According to Fröhlich and Madeo, 2000 [6], yeast cells undergoing apoptosis display characteristic markers as DNA and mitochondria fragmentation, reactive oxygen species (ROS) accumulation or cytochrome *c* release. The discovery of apoptosis in yeasts has suggested that other forms of RCD might occur, such as programmed necrosis, an active regulatory mechanism taking place along yeast chronological aging [2,5]. Studies by Orozco et al. (2013) [7] observed that factors associated to apoptosis, such as the caspase Mca1p and the apoptosis-inducing factor Aif1p, play a positive role in yeast longevity during winemaking. More recently, results obtained by Duc et al. (2017) [8] indicated that yeast cell death results from its inability to trigger an appropriate stress response under some conditions of nutrient limitations. 

Once yeast cells die, the yeast intracellular content is released into the wine due to the action of hydrolytic enzymes during autolysis, an irreversible lytic event associated with cell death [9,10]. This process has been reported to take place during the aging period in sparkling wines elaboration, which is essential for wines to obtain its characteristic properties [11,12]. In order to provide the expected results, yeast cells are subjected to long aging periods and low temperatures that lead to eventual autolysis [13]. While this step is crucial in the process of sparkling wine elaboration, the time restraints make it a bottleneck and consequently, choosing strains with accelerated autolysis is a desirable feature and it has become an object of study of numerous researchers [14,15,16].

One such strain that could be a candidate for this process are flor yeasts. These yeasts tend to form biofilms on the wine surface under nutrient limitation conditions (among other stresses) during biological aging in sherry wines production [17,18,19,20]. Therefore, the benefit of a cellular suicide program seems logical, since apoptosis might increase the nutrients availability for younger cells through the cell death of the older and unhealthy members from the yeast population [6]. However, while autolysis process has been evidenced under enological conditions [21], apoptosis remains unreported in flor yeast so far, although it should be noted that apoptotic factors have been observed during biofilm formation [22].

In this study, the apoptosis and autolysis-related proteome of two industrial yeast strains used commonly in post-fermentative processes (sparkling wine secondary fermentation and biological aging), was analyzed and compared in order to detect potential biomarkers under CO_2_ overpressure conditions (typical of sparkling wine elaboration process).

## 2. Materials and Methods

### 2.1. Yeast Strains

Two industrial *S. cerevisiae* yeast strains, which are used after a primary fermentation, were compared in this study: a yeast strain typically used in sparkling wines’ fermentation process (P29 CECT 11770), isolated from INCAVI (Catalan Institute of Vines and Wines); and a flor yeast strain (G1 ATCC MYA-2451) isolated from a wine velum from PDO Montilla-Moriles (Spain) with 14.5% *v*/*v* ethanol content, an unconventional yeast strain for sparkling wine elaboration responsible for biological aging of sherry-wines.

### 2.2. Second Fermentation Conditions and Sampling Times

Yeast cells were previously grown in YPD medium at 21 °C for 48 h. Later, each strain was inoculated (1.5 × 10^6^ cells/mL) separately in a pasteurized grape juice (Macabeo white grape variety with 174.9 g/L of sugar, 3.6 g/L total acidity and pH 3.4) as acclimation medium and maintained with gentle stirring (100 rpm) at 22 °C for 5 days. Once yeast cells reached high values of cell concentration (1.5 × 10^8^), viability (97%) and ethanol (10.39% *v*/*v*), the must was properly filtered (using a sterilized pore filter of 0.45 μm) and “tirage” was carried out using standard sparkling wine bottles of 750 mL, according to INCAVI protocol.

A total of 12 bottles were used for each strain (24 in total) and filled with the fermenting mixture: a standardized commercial base wine (Macabeo and Chardonnay 6:4, 10.21% *v*/*v*, 0.3 g/L of sugar, pH 3.29, 5.4 g/L of total acidity and 0.21 g/L of volatile acidity), 22 g/L of sucrose and 1.5 × 10^6^ cells/mL. Half of the bottles (six) were closed with a bidule (plastic lid) and a metal overcap, representing the endogenous CO_2_ overpressure condition (PC); and the rest of the bottles (six) were closed with a perforated bidule to allow the output of CO_2_, representing the non-pressure condition (NPC). Bottles under both conditions were incubated at 14 °C. Kinetics of second fermentation by both strains was performed measuring the pressure values with an internal aphrometer (Oenotilus, Station Oenotechnique de Champagne, Epernay, France). Samples of each strain were taken from three random bottles at two times during the second fermentation: At the middle, when pressure reached 3 bars; and 1 month after it, once the maximum pressure value was measured (6.5 bars). At the same time, samples from the control bottles without pressure were collected considering similar values under both conditions of sugar consumption (T1: 9.07 ± 0.26 g/L and T2: 0.3 ± 0.0 g/L) and ethanol content (T1: 10.74 ± 0.03% *v*/*v* and T2: 11.56 ± 0.04% *v*/*v*).

### 2.3. Viability

Viable yeast cells counting was carried out using appropriate dilutions with Ringer solution. These were then plated in Sabouraud agar medium for 48 h at 28 °C and all samples were analyzed by triplicate.

### 2.4. General Wine Parameters

Ethanol content (%, *v*/*v*), reducing sugars, volatile and total acidity, and pH were quantified according to methods described by OIV (2018) [23]. Malic and lactic acid were analyzed using enzymatic method through a multi-parametric analyzer Lisa 200 (Hycel Diagnostics, Technology Diffusion Iberica, Barcelona, Spain).

### 2.5. Protein Extraction, Identification and Quantification

Methods described in Moreno-García et al. (2015) [24] and Ishihama et al. (2005) [25] for protein extraction and identification, and quantification were used, respectively. Yeast cells from each strain were collected separately by centrifugation and lysed using a Vibrogen Cell Mill. Proteins were extracted and separated according to their isoelectric point using the OFFGEL High Resolution kit pH 3–10 (Agilent Technologies Palo Alto, CA). These proteins were digested into peptides with trypsin and identified through mass spectrometry using a LTQ Orbitrap XL (Thermo Fisher Scientific, San José, CA, USA) coupled to a nano LC Ultimate 3000 system (Dionex, Germany).

Once proteins were identified, apoptosis and autolysis-related proteins were selected through the Gene Ontology section from the *Saccharomyces* genome database (SGD, http://www.yeastgenome.org/) and Uniprot (http://www.uniprot.org/) databases.

### 2.6. Confidence Parameters and Statistics

Proteins detected with score > 2 and observed peptides ≥ 2 were selected to proceed with the analysis. These parameters are provided in Appendix A for each process. Regarding the most abundant common proteins, those whose content ratio PC/NPC reached values ≥ 2 (over-represented) and ≤ 0.5 (down-represented), as well as the specific proteins of each strain, were highlighted and discussed in detail. Moreover, proteins found with the highest contents were considered as well. Among the protein group annotation, *p*-values as well as the FDR or False Discovery Rate for each one, were calculated by the tool “GO Term finder” from SGD database. The total of ORFs identified in *S. cerevisiae* (according to the SGD database) was used as a background set of genes and it was taken into account only by those genes which are translated into proteins. GO terms with *p*-values lower than 0.01 were highlighted. The software Statgraphics Centurion v.XVI was used to perform a multiple-sample comparison (MSC) with a confidence level of 95.0%, according to Fisher’s least significant difference (LSD) method, One-way ANOVA and Principal Component Analysis (PCA). The software STRING (version 11.0) was used to create the interaction network map. Correlation analysis was performed using the version 4.0 of the software MetaboAnalyst. Data were properly normalized (square root) and auto scaled, prior to analysis.

## 3. Results and Discussion

### 3.1. Second Fermentation Kinetics and Cell Viability

Figure 1 shows the evolution of endogenous CO_2_ released by *S. cerevisiae* P29 and G1 along the second fermentation. The sparkling wine yeast strain P29, commonly used in the elaboration of these wines, reached 3.3 bars at 8 days and 6.5 bars at 23 days, representing the middle and the end of second fermentation, respectively. On the other hand, flor yeast G1 showed a slightly slower kinetics, obtaining 3.6 bars at 10 days, while the maximum value of 6.4 bars was measured at 28 days. The wine composition at this final point in both strains was as follows: 11.56 ± 0.04% *v*/*v* ethanol, 0.3 ± 0.0 g/L reducing sugars, 0.23 ± 0.02 g/L volatile acidity, 5.25 ± 0.02 g/L total acidity, 3.3 ± 0.02 pH, 1.89 ± 0.06 g/L malic acid, 0.1 ± 0.0 g/L lactic acid in *S. cerevisiae* P29; and 11.4 ± 0.2% *v*/*v* ethanol, 0.3 ± 0.0 g/L reducing sugars, 0.28 ± 0.01 g/L volatile acidity, 4.3 ± 0.3 g/L total acidity, 3.37 ± 0.01 pH, 0.2 ± 0.1 g/L malic acid, 1.2 ± 0.1 g/L lactic acid in *S. cerevisiae* G1.

As it is observed in Figure 2, cell viability decreased considerably over time and it was more pronounced and significant under PC. Focusing on each strain separately, remarkable differences can be noticed in samples subjected to CO_2_ pressure (PCT1 and PCT2), where the yeast strain P29 outperformed G1, reaching more than double in both sampling times. In addition, P29 also obtained high viability values under NPC, especially at T1.

Second fermentation kinetics and viability results reveal that apart from enduring better pressure, *S. cerevisiae* P29 is also more adapted to the medium conditions, which may be logical since it is typically used during sparkling wine elaboration. Nevertheless, it is not the case of G1, whose kinetics and viability seem to be affected negatively by these factors. These differences observed in both yeast strains could be explained due to the different environmental conditions, in terms of ethanol content, pressure or nutrients availability, in which yeast cells have to cope along with the prise de mousse. According to the results, the study of cell death-related proteome in these yeast strains would be of great interest to try to characterize their response under pressure conditions.

### 3.2. Study of the Proteome

In this work, the total of proteins identified after analysis, in both *S. cerevisiae* P29 and G1, was: 594 and 568, respectively, under PCT1; 1517 and 1000 under NPCT1; 419 and 94 under PCT2; 392 and 218 under NPCT2. In *S. cerevisiae*, a total of 24 and 184 proteins related to apoptosis and autolysis, respectively, have been reported so far, representing 0.36 and 2.79% of the total proteome (6604 ORFs, according to SGD database). From the proteomes associated with these processes, 13 (54.17%) and 84 (45.65%) apoptosis and autolysis proteins, respectively, were found in this study. The frequency values were calculated considering the total proteome identified under each condition, as mentioned above. It highlighted the high frequency values obtained in both strains for apoptosis proteins (2.30% in P29 and 2.29% in G1) under NPCT2. As for the total autolysis-related proteins frequencies, the value observed in G1 under NPCT2 (6.88%) stood out among the rest. Hydrolytic enzymes, mainly glucanases and vacuolar proteases, were relevant in terms of frequency in flor yeast G1 under PCT2. On the contrary, non-vacuolar proteases were more abundant at T1 under both conditions and especially in G1 (3.70% under PCT1 and 3.30% under NPCT1).

Regarding the total number of proteins, P29 showed a greater abundance of both apoptosis and autolysis in all sampling times, except under PCT1 for apoptosis, and PCT1 and NPCT2 for autolysis. Nevertheless, this increase is not reflected in terms of content. Total values of protein content (mol%), provided in Table 1, were higher in flor yeast G1 than in P29, considering the two processes, except in the case of apoptosis under NPCT2 and autolysis under NPCT1. Differences in content of apoptosis proteins between strains were established under each condition, except under NPCT2 (Table 1). On the contrary, pressure only boosted the content of autolysis proteins in flor yeast, even though the highest protein content was reported under NPCT2 (Table 1). As it is observed, differences between yeast strains, considering the autolysis proteome, only appeared once the maximum value of pressure (6.5 bars) was reached at T2. These results suggest that both CO_2_ overpressure and fermentation stage affect flor yeast, especially in the apoptosis-related proteome, which may be related to the decrease in viability experimented in sealed bottle at the first month of aging (Figure 2). Focusing on the autolysis enzymes, a similar pattern was observed. Along the second fermentation, the content of glucanases and vacuolar proteases increased more under PC in flor yeast (Table 1), whereas nucleases in general were more abundant in P29. The presence of glucanases and vacuolar proteases with high content in flor yeast at T2 may be explained due to the structural and morphological changes during autolysis [26]. Since this yeast strain is not adapted to pressure conditions, the requirement of proteins involved in cell wall remodeling could be more relevant in flor yeast than in P29. This result, along with the high content of apoptosis proteins reported, could make flor yeast interesting from an industrial and enological point of view, thus favoring a faster cell death and later compounds release during aging. Apart from vacuolar proteases, those not located in this organelle were the most remarkable in terms of protein number at T1, although this value dropped at T2 and it was more noticeable in the case of flor yeast. Moreover, the content of these enzymes seems to be affected by fermentation stage, decreasing from T1 to T2 (Table 1), rather than pressure. On the other hand, mannosidases were not observed under PC in flor yeast, only in P29 at T2, indicating a negative effect of pressure.

In order to observe the possible protein–protein connections between apoptosis and autolysis-related proteins in both strains, interaction maps built using the STRING v.11.0 database are provided in Figure 3 and Figure 4. The connections between proteins are established by edges and the edges thickness represents the strength of these connections. From the total of 13 apoptosis proteins in both strains, only 17 interactions between them were reported, whereas more numbers of edges (350) and stronger connections were established in the case of autolysis proteins. The connections between proteins were obtained with a p-value < 6.21 × 10^−10^ for apoptosis and < 1 × 10^−16^ in the case of autolysis proteins. This enrichment indicates that the proteins are partially biologically connected as a group. Moreover, proteins were grouped into different clusters through MCL (Markov Cluster Algorithm) clustering method. Apoptosis proteins were sorted into three different clusters (Figure 3), represented by red, green and blue nodes. Connections between the proteins Nma111p, Mca1p and Bir1p (red nodes) and Tdh2p, Tdh3p, Por1p and Fis1p (green nodes), were reported as the strongest, while Cpr3p was the only protein that showed no interaction with the rest. On the contrary, a higher number of cluster and nodes were observed in the autolysis map (Figure 4). The red nodes were perfectly separated in the map, most of them belonging to the proteasome subunits. Among them, the alpha and beta subunits (Pre1-10p and Pup1-3p) showed the strongest interactions considering all the autolysis proteins identified. Apart from these, the ubiquitin-dependent proteins (Ubp1-3p, Ubp6p, Ubp7p, Ubp12p, Ubp14p, Ubp15p, Ulp2p, Otu1p, Rex2p, Rpn11p, and Yuh1p) are also highlighted in the red cluster. This could indicate a role of the proteasome complex and ubiquitin-dependent catabolism during autolysis, although it has been not reported so far. In addition, no interactions were reported for the protein Yil108wp.

To achieve a more detailed conclusion, each process as well as the most remarkable proteins identified (over and down-represented, specific proteins of each strain and those showing high protein content, all of them under PC) have been discussed separately.

1.  Apoptosis

Apoptosis-related proteins identified in both yeast strains are provided in Table 2. The most remarkable apoptosis-related proteins are shown in Table 3 where those that were over-represented under PC were: Cpr3p, Esp1p, Nma111p and Oye2p in G1 at T1; whereas at T2, only Oye2p was over-represented in P29.

Oye2p, over-represented three-fold under PCT1 in flor yeast (Table 3), is a conserved NADPH oxidoreductase whose overexpression has been demonstrated to lower endogenous reactive oxygen species (ROS), increase resistance to H_2_O_2_-induced PCD, and significantly decrease ROS levels generated by organic prooxidants [27]. On the other hand, the fact that Oye2p is over-represented in P29, only under PCT2, can be associated with ethanol stress response [28]. The cyclophilin Cpr3p, located in the matrix of yeast mitochondria and involved in protein folding [29], has been observed to mediate Cu-induced apoptosis in yeast [30] and some forms of necrotic death [31,32]. Additionally, proteins such as Esp1p and Nma111p were found in flor yeast only under PCT1, while in P29 they appeared exclusively under NPCT1. The caspase-like protease Esp1p has been associated with the apoptosis promotion in budding yeast, through the cleavage of the protein Mcd1p [33]; whereas the interaction of the nuclear mediator of apoptosis Nma111p and Bir1p (Figure 3) has been reported to induce apoptosis [34]. The high content under PCT1 of these proteins in flor yeast may be associated with the raise of typical stresses of sparkling wine production [35], conditions which are not common of this type of yeasts. Apart from these stresses, the absence of oxygen in a sealed bottle may compromise the viability of flor yeasts, since under oxidative conditions these yeasts are known to form biofilm, which allow their growth on non-fermentable carbon sources such as ethanol, assuring cell survival under such conditions [17,18].

As for the rest of the proteins, although they were not found to be over-represented under PC, the GAPDH (glyceraldehyde-3-phosphate dehydrogenase) isoenzymes Tdh2p and Tdh3p highlighted in flor yeast under PCT2 in terms of content, two-folding the P29 content (Table 2). Although these enzymes have been reported to be crucial mediators of yeast H_2_O_2_-induced apoptosis together with nitric oxide (NO) [36] and a specific substrate of yeast metacaspase [37], they are known to be essential glycolytic enzymes [38]. Therefore, the difference in content of these enzymes may be due to a different metabolic response to environmental conditions. The metacaspase Mca1p was down-represented under PC in both strains at T1, and at T2 was only detected in P29 (Table 2), even though its protein content did not highlight. Furthermore, the RNase Rny1p, was also found to be down-represented under PCT1 in both strains, although in this case its content at T2 was higher than the caspase. Thompson and Parker (2009) [39] showed that in yeast, tRNAs are cleaved by this protein and released from the vacuole into the cytosol during oxidative stress, promoting apoptosis. Other interesting proteins identified in both strains to be down-represented under PC were: Por1p, a mitochondrial outer membrane protein porin 1 reported to regulate negatively acetic acid-induced apoptosis by an AAC (ADP/ATP carrier)-dependent mechanism [40]; Oye3p, which has an opposite role to Oye2p [27]; Fis1p or mitochondrial fission protein, found exclusively in P29, thought to mediate both ethanol-induced apoptosis and mitochondrial fragmentation [41], although without correlation with cell death; and Bir1p, member of the apoptosis inhibitor family involved in cell division regulation [42]. Moreover, it should be noted that the Aif1p or apoptosis-inducing factor, known to be translocated to the nucleus in response to apoptotic stimuli [43], was detected only in flor yeast under NPCT2 with a low content, but it was not selected to proceed with the proteomic analysis since it was detected with a score value and peptides < 2.

As it has been mentioned above, a significant drop in viability was observed under pressure conditions along the second fermentation (Figure 2). Therefore, those proteins detected under this condition may represent possible cell death biomarkers. In order to analyze this relationship, the software MetaboAnalyst was used to perform a correlation matrix between the content of apoptosis-related proteins and viability values identified just in a sealed bottle for each strain (Figure 5). Clustering analysis separated the proteins in two groups according to the type of correlation with cell viability. In the case of apoptosis in P29 (Figure 5A), most of the proteins showed negative correlations: Mca1p (p-value: 4.7 × 10^−5^), Oye2p (1.8 × 10^−5^), Por1p (4.7 × 10^−5^), Rny1p (4.7 × 10^−5^), Tdh2p (1.5 × 10^−5^), and Tdh3p (1.7 × 10^−5^). On the contrary, those in G1 (Figure 5B) showed both positive: Oye2p (3.7 × 10^−5^), Cpr3p (3.6 × 10^−5^), Nma111p (3.4 × 10^−5^), and Esp1p (2.8 × 10^−5^); and negative: Tdh2p (5.7 × 10^−5^) and Tdh3p (6.1 × 10^−5^). All these proteins correlated with viability under PC may represent possible biomarkers during the second fermentation, since as it has been mentioned above, all of them have been related to cell death [37,39,40] or stress response [27]. Moreover, those proteins are also found to be over-represented under PC such as Oye2p in both strains; and Cpr3p, Esp1p and Nma111p just in G1, could be proper biomarkers in order to accelerate cell death and promote further autolysis in these wine yeasts [14,15].

2.  Autolysis

Glucanases, nucleases and proteases, especially those located in the vacuole, are the main hydrolytic enzymes involved in yeast autolysis. Autolysis-related proteins identified in both yeast strains are provided in Appendix A. In these tables, relevant proteins in terms of content and those significant are highlighted. Among the GO annotations, “proteolysis” was the most significant biological process in both strains during autolysis, especially under NPCT1, followed by others related to proteasome functions and the organonitrogen compound metabolic process (Appendix A).

During yeast autolysis, glucanases and mannosidases were reported as the main enzymes involved in cell wall degradation [10]. Among the glucanases over-represented under PC, only the exo-1,3-beta-glucanase Exg2p involved in cell wall beta-glucan assembly was detected in G1 at T1 (Table 3). Bgl2p and Exg1p (not over-represented under PC) were found with the highest contents in both strains, especially in flor yeast (Appendix A). These endo and exo-beta-1,3-glucanases, respectively, are known to be the major glucanases of the cell wall required for cell wall maintenance and incorporation of newly synthetized mannoprotein molecules into the cell wall and beta-glucan assembly [44,45]. The presence of both over-represented proteins (Exg2p) and those most abundant under PC (Bgl2p and Exg1p) may indicate higher glucans and mannoproteins degradation and release during autolysis. Nevertheless, mannosidases were found with low protein content in both yeast strains and only the vacuolar mannosidase Ams1p involved in free oligosaccharide degradation [46] was detected to be over-represented under PC T2 only in P29 (Table 3).

Regarding those proteases not located in the vacuole, differences in terms of protein number were observed in the over-represented proteins under PC, depending on the yeast strain. Whereas in flor yeast, these over-represented proteins were found only at T1; in the case of P29, they were specifically detected at T2 (Table 3). In flor yeast, the following stood out: the leucine aminopeptidase Ape2p involved in peptides metabolism [47], which obtained a value 3.34-fold higher under PCT1; the protein Mas2, detected 3.25-fold under PC and required for mitochondrial processing protease (MPP) [48]; and the cysteine aminopeptidase Lap3p [49]. The presence of these proteins over-represented in flor yeast may be explained due to their function in protein and peptides processing, thus promoting amino acid and peptide release during autolysis [12]. Apart from these, other proteases mentioned above (Esp1p and Nma111p), proteasome subunits (Pre4p, Pre10p and Ubp6p) and a metalloendopeptidase (Ste24p) were also found to be highly represented under PC or specifically in flor yeast. On the other hand, in P29, only three proteins (Pre6p, Pre7p and Rpn11p) were found to be over-represented under PCT2 and specific to this strain (Table 3). As in flor yeast, most of the proteins identified in P29 to be over-represented were involved in proteasomal functions, which is in accordance with the GO terms associated with proteasome in both strains (Appendix A). The 26S proteasome, which comprised of a 19S regulatory particle and a 20S catalytic core particle, is the protease responsible for the non-vacuolar degradation of cellular proteins [50,51] containing 14 subunits (seven alpha-type and seven beta-type). All these proteasome subunits along with those involved in ubiquitin-dependent catabolism as Ubp proteins, were perfectly clustered (red nodes) in the interaction map (Figure 4) and indeed, they showed the strongest connections among the total. These results suggest an important role of this complex during autolysis, although it has not been reported yet. Additionally, other proteins such as the peptidase Dug1p, involved in glutathione degradation [52], were not detected as over-represented under PC, but reached a maximum value in flor yeast at NPC T2 (Appendix A), which might explain the fact that at this time and condition in this strain, the total content of peptidases experimented with an increase, as was mentioned above. Also, this protein was identified under PC T2 specifically in P29 (Appendix A).

Yeast vacuole contains a total of seven proteases (reported thus far) which contribute to a wide range of essential functions, being the principal function the protein degradation under starvation conditions [53]. In fact, Teichert et al. (1989) [54] observed a degradation of up to 85% of the intracellular content during this stress. These vacuolar proteases include three aminopeptidases (Ape1p, Ape3p and Dap2p), two carboxypeptidases (Prc1p and Cps1p) and two endopeptidases (Pep4p and Prb1p), even though recently, Parzych et al. (2018) [55] reported the presence of another carboxypeptidase, Atg42p/Ybr139wp, required for normal vacuole function and autophagy. Among the vacuolar proteases, only Ape1p and Ybr139wp were found to be over-represented under PC at T1, in the case of flor yeast, whereas no proteins were highly represented at T2 (Table 3). Both the aminopeptidase Ape1p and the carboxypeptidase Ybr139wp have been involved in autophagy [55,56,57]. Moreover, even though it was not found to be over-represented, the proteinase A or Pep4p was identified in all the sampling times and it reached the highest protein content in both strains, standing out in flor yeast under PCT2 (Appendix A). This vacuolar protease has been demonstrated to be the main enzyme involved in yeast autolysis and responsible for 80% of the nitrogen release [58]. In addition, a protective role has been observed during acetic acid-induced apoptosis [59] and chronological aging [60]. As for the rest of vacuolar proteases, it highlighted the carboxypeptidase Prc1p [61], which obtained high values of protein content in flor yeast and especially at T2 (Appendix A). Finally, in addition to proteases and glucanases, nucleases involvement in autolysis has been studied under enological conditions [62,63], but further studies are necessary to clarify the impact on wine quality. In this study, over-represented nucleases under PC were found only at T1 in both strains. Among them, the endonuclease Vma1p [64] highlighted in flor yeast and obtained a value two-fold higher under this condition (Appendix A). Moreover, other nucleases such as Ybl055cp were also found to be specific to flor yeast. According to Qiu et al. (2005) [65], this protein may participate in DNA degradation during apoptosis. In P29, however, only Dna2p involved in DNA elongation [66] was over-represented under PC (Table 3).

In addition to the flocculation capacity reported in flor yeast [18] and its ethanol tolerance, an accelerated autolysis represents a desirable feature for the sparkling wines production, since it would allow a faster cell compound release and acquisition of the organoleptic properties during aging. The detection of both glucanases and vacuolar proteases highly represented in sealed bottle conditions and others with high content in flor yeast, make this type of yeast an interesting alternative for sparkling wine improvement and elaboration. In fact, the selection of flocculent yeasts and the study of the autolytic ability of these yeasts to improve sparkling wine fermentation have been reported [67,68,69]. Furthermore, the impact of flor yeasts on the sparkling wine quality has been recently observed [70]. These authors used for the first time a flor yeast to carry out a second fermentation in a bottle and revealed that aroma compounds are affected by pressure and fermentation stage.

3.  Principal Component Analysis (PCA)

Those proteins showing between 4–8 homogeneous groups (HG), at 0.05 level, according to Fisher’s least significant difference procedure (highlighted with an asterisk in Table 2 and Appendix A) in both yeast strains and biological process, were subjected to a Principal Component Analysis (PCA) (Figure 6). This analysis allows for the establishment of linear combinations between the different variables which account for most of the variability of the data and provide classification among the samples.

Three components whose values are greater than or equal to 1.0, have been extracted in both PCAs. In this way, the three PCs obtained for the biplot of apoptosis proteins explain 84.01% of the total cumulative variance by the eight selected variables. PC1 accounts for 39.21% of the variance, while PC2 and PC3 explain 28.02 and 16.79%, respectively. According to the results, PC1 seems to distinguish between yeast strains, locating the P29 and G1 samples on the right and left side, respectively. The contribution of each protein to each component is established by their coordinate values on the corresponding axis. Proteins such as Rny1p (0.51) and Mca1p (0.49) in P29, and the GAPDH proteins Tdh2p (−0.27) and Tdh3p (−0.25) in G1, were those that contributed the most to this difference between strains. On the other hand, PC2 grouped the samples based on the time effect, establishing a separation between T1 samples (on the left with negative scores) and those at T2 (on the right with positive scores). This component was mainly influenced by Tdh2p (0.54) and Tdh3p (0.52) at T2, and Oye2p (−0.33) at T1. Differences in time may be explained since cell death was more relevant at the end of the fermentative process, as is shown in Figure 2 for the viability, which may be caused by the pressure effect. On the contrary, apoptosis does not seem to be affected at T1, since samples of both strains were perfectly grouped. Last, PC3 differentiated the samples depending on the condition: PC samples seem to be located on the positive side of the component, while the control samples NPC are on the negative side. This component correlates mainly with the content of Oye2p (0.22 and −0.72) and the GAPDH enzymes (both with 0.22). The pressure effect is more remarkable in the samples taken at T2, and especially in flor yeast G1, whose samples show more separation, which correlates with the higher drop in viability at this last time under PC (Figure 2).

As for the biplot of autolysis proteins, the three PCs account for 85.47% of the total variance. PC1 explains 40.18%, PC2 29.45% and the last component PC3, 15.84%. PC1 clearly grouped the samples taken at T1 (negative scores) from those at T2 (positive scores). It was mainly influenced by the endonuclease Vma1p (−0.32) and proteasome subunits as Pre8p and Pre9p (both −0.32) at T1, whereas glucanases Exg1p (0.36) and Bgl2p (0.26), as well as the vacuolar protease Pep4p (0.36) at T2. This difference could indicate an important role of proteasome at the middle of the second fermentation and a high requirement of proteins involved in cell wall remodeling during the first month of aging, which also participate in cell wall degradation and peptides release during autolysis. Moreover, this process is known to take place 2–3 months after the end of second fermentation [10,12,58]. PC2 seems to differentiate between P29 samples taken at T1, on the left, from those at T2, on the right, and correlates with proteasome subunits (mainly Pup3p: −0.32 and Pre5p: −0.29) at P29T1, and Rny1p (0.39) and Mca1p (0.37) at P29T2. On the other hand, PC3 appears to establish two sample groups. The first one located on the negative side of PC3, grouping the G1 samples under NPCT1 and PCT2, and the other on the positive side contains the samples G1NPCPT2 and G1PCT1. The proteins Pre8p (−0.10), Dug1p (0.55), and the aminopeptidases Ape1p (0.51) and Ape2p (0.50) were those that contributed the most to this component. As it is observed, the pressure effect on autolysis-related proteome in both strains is more noticeable in samples obtained at the first month of aging, which is the time when autolysis starts. In addition, this effect seems to be more relevant in G1 whose samples showed a higher separation than those from P29.

## 4. Conclusions

In this work, a proteomic analysis was carried out in order to detect possible cell death biomarkers in yeast through the study of apoptosis and autolysis proteome along the secondary fermentation in two industrial strains. This analysis was carried out in two yeast strains used in post-fermentative processes. All results considered, the presence of apoptosis and autolysis proteins could show evidences that these processes are taking place, and an effect of CO_2_ overpressure on the cell death-related proteome is observed. Additionally, over-represented proteins and those correlated with cell viability under pressure conditions such as Cpr3p, Esp1p, Oye2p, and Nma111p, as well as proteins detected non strain-dependent as Bgl2p, Exg1p, and Pep4p, may be proper biomarkers in order to accelerate cell death and improve yeast autolytic capacity during aging in sparkling wine elaboration. In addition, the use of multivariate statistical analysis such as PCA, showed differences in the behavior of apoptosis and autolysis proteome and allowed to group the samples according to the effect of the different factors studied: strain, time and pressure. The differentiation of the samples seems to be due to both pressure and fermentation stage effect.

From the industrial and enological point of view, the use of a flor yeast in sparkling wine production and improvement may be considered. Apart from its ethanol tolerance and flocculation ability, the fast cell death observed along the prise de mousse, its high content in glucanases and vacuolar proteases, as well as the numerous highly-represented proteins under pressure conditions, make this strain an interesting alternative within the sparkling wines field. However, further proteomic and metabolomics studies focused on the organoleptic properties of flor yeasts, genetic approaches, and electronic microscopy observations are required in order to achieve more solid conclusions. The proteomic profiles obtained in this study could be considered at the time of selecting wine yeast strains for sparkling wine production and aim to shed light on the knowledge of yeast behavior when these are in a life and death debate under enological conditions.

## Figures and Tables

**Figure 1 microorganisms-07-00542-f001:**
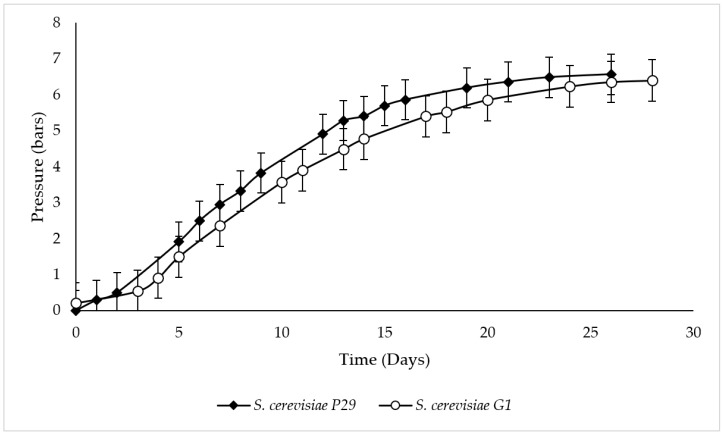
Kinetics of second fermentation performed by *S. cerevisiae* P29 and G1.

**Figure 2 microorganisms-07-00542-f002:**
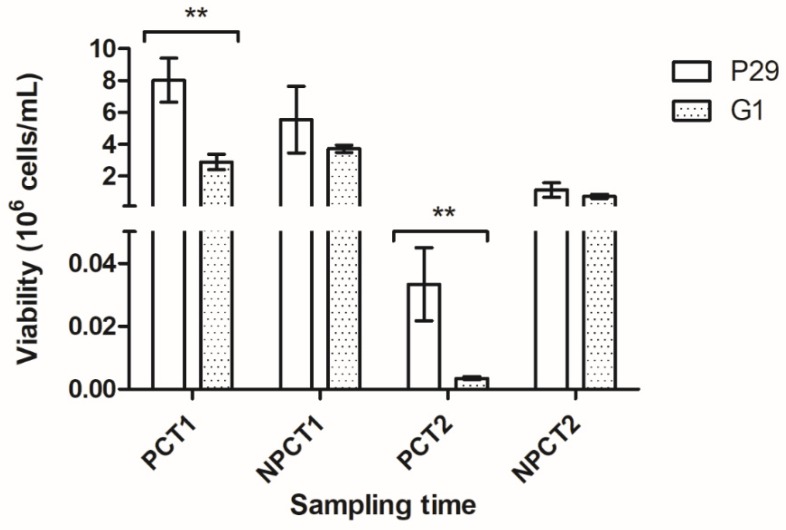
Cell viability (cells/mL) observed in both *S. cerevisiae* yeast strains P29 and G1 under PC (CO_2_ pressure condition) and NPC (non-CO_2_ pressure condition), in each sampling time: at the middle of the secondary fermentation (T1) and 1 month after the secondary fermentation (T2). Error bars represent the standard deviation of the three independent experiments. Significant differences from One-way ANOVA analysis are indicated by *, depending on the significance level (** < 0.01).

**Figure 3 microorganisms-07-00542-f003:**
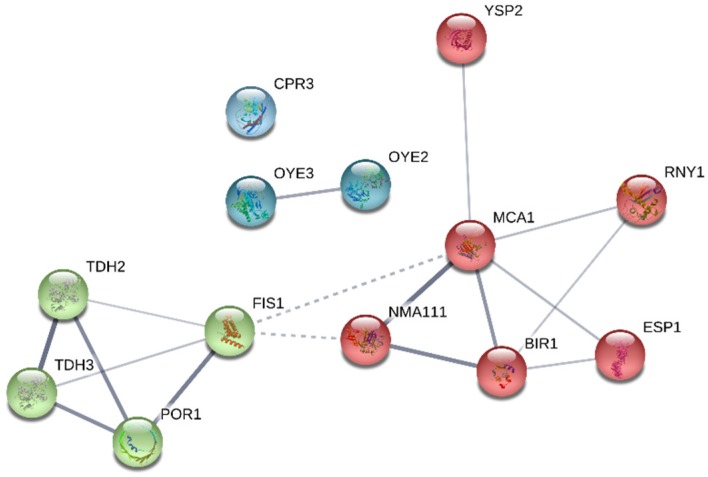
Interaction network map built using STRING v11.0 and based on the 13 apoptosis-related proteins in total detected in *S. cerevisiae* P29 and G1. Proteins are shown as nodes and the existence of interactions between them are represented by edges (connection between nodes). Edges thickness indicates the strength of the different interactions. Nodes with the same color represent specific clusters. PPI enrichment *p*-value < 6.21 × 10^−10^.

**Figure 4 microorganisms-07-00542-f004:**
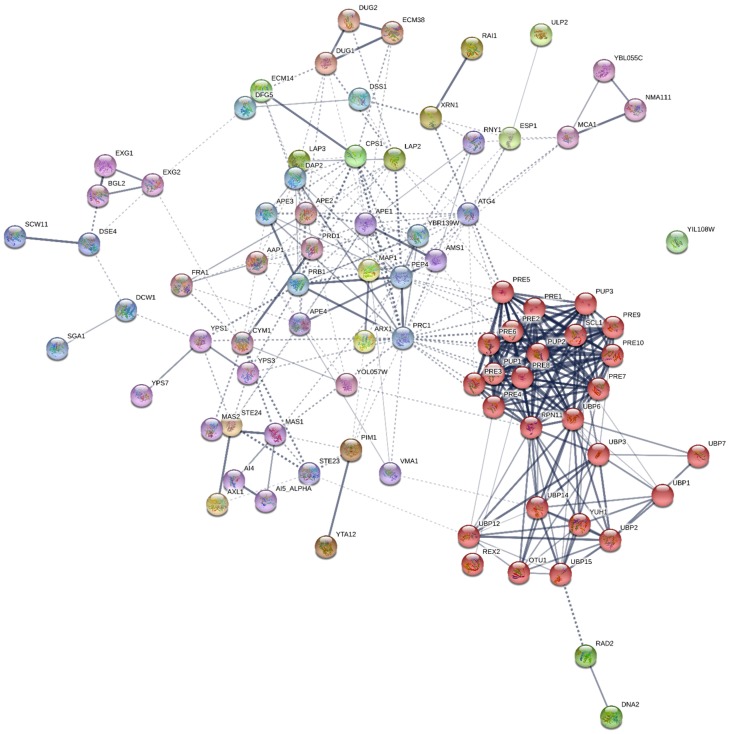
Interaction network map built using STRING v11.0 and based on the 84 autolysis-related proteins in total detected in *S. cerevisiae* P29 and G1. Proteins are shown as nodes and the existence of interactions between them are represented by edges (connection between nodes). Edges thickness indicates the strength of the different interactions. Nodes with the same color represent specific clusters. PPI enrichment *p*-value < 1 × 10^−16^.

**Figure 5 microorganisms-07-00542-f005:**
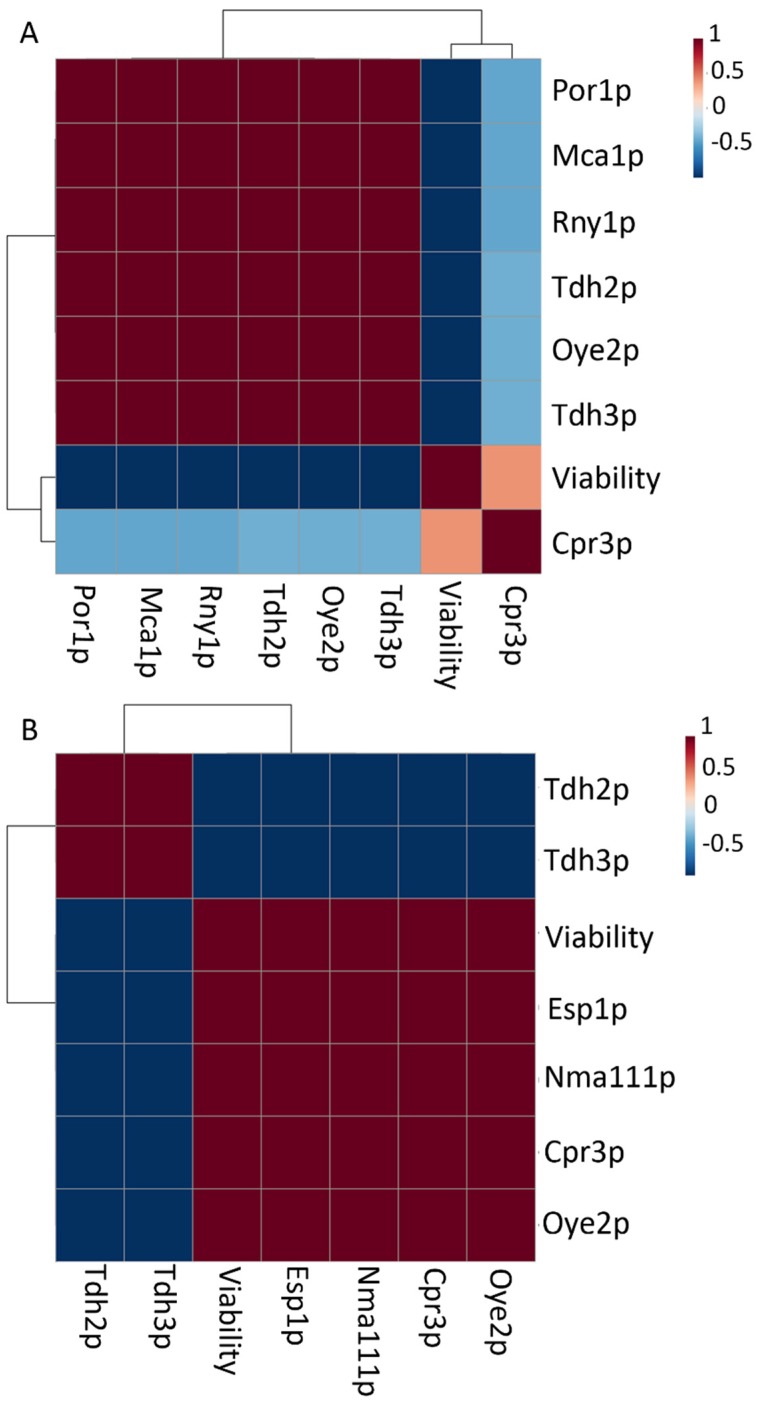
Correlation matrix and clustering analysis based on the content of normalized and scaled data of apoptosis proteins and cell viability observed under CO_2_ overpressure conditions in: (**A**) *S. cerevisiae* P29 and (**B**) G1.

**Figure 6 microorganisms-07-00542-f006:**
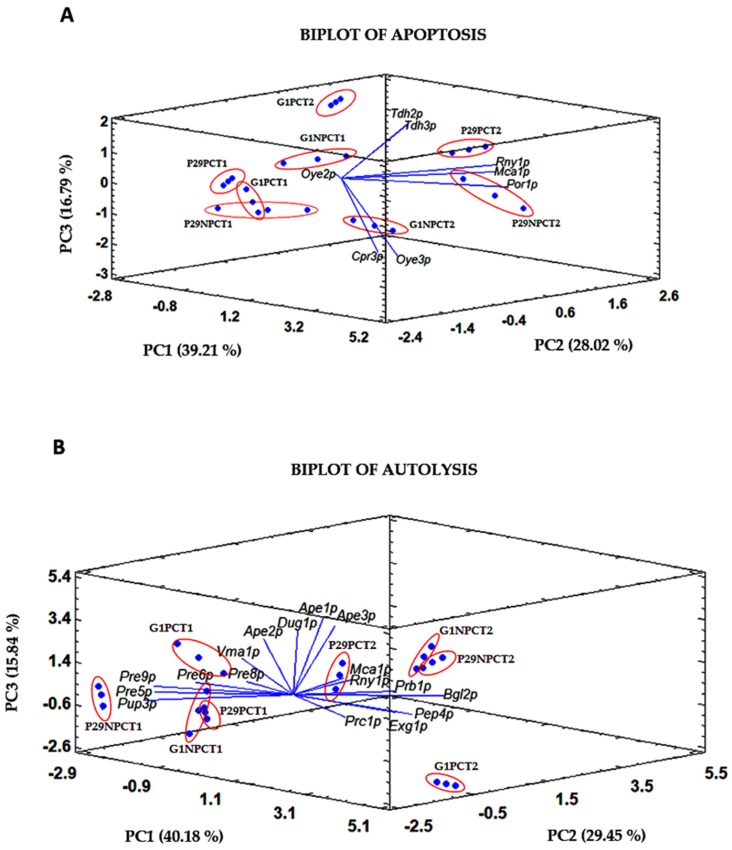
Biplot of Principal Component Analysis (PCA) based on the significantly different proteins identified in *S. cerevisiae* P29 and G1, related to apoptosis (**A**) and autolysis (**B**). Samples are shown in read circles and replicate values in blue dots.

**Table 1 microorganisms-07-00542-t001:** Total average value of protein content (mol%) of the proteins related to apoptosis and autolysis identified in *S. cerevisiae* P29 and G1 under pressure (PC) and non-pressure (NPC) conditions, and in each sampling time: at the middle of the secondary fermentation (T1) and 1 month after it (T2). Total proteome identified so far in *S. cerevisiae* is provided. A–e Different letters in a row represent significant differences of protein content in each condition at 0.05 level according to Fisher’s least significant difference procedure. n.f. not found.

Condition	PC T1		NPC T1		PC T2		NPC T2	
Yeast Strain	P29	G1	P29	G1	P29	G1	P29	G1
	mol% ± SD	mol% ± SD	mol% ± SD	mol% ± SD	mol% ± SD	mol% ± SD	mol% ± SD	mol% ± SD
**Total Apoptosis**	1.7b ± 0.2	2.5c ± 0.3	1.2a ± 0.2	2.7c ± 0.2	3.2d ± 0.3	4.5e ± 0.2	3.4d ± 0.4	3.5d ± 0.3
**Total Autolysis**	3.3a ± 0.4	3.7a ± 0.7	5bc ± 1	3.3a ± 0.8	3.3a ± 0.5	4.9bc ± 0.3	3.7ab ± 0.4	5.9c ± 0.5
*Glucanases*	0.73b ± 0.08	0.51a ± 0.04	0.7b ± 0.2	0.6ab ± 0.1	1.2c ± 0.2	2.4e ± 0.1	1.7d ± 0.2	2.18e ± 0.09
*Mannosidases*	n.f.a	n.f.a	0.13c ± 0.05	0.05b ± 0.02	0.022b ± 0.001	n.f.a	n.f.a	0.104c ± 0.001
*Non-vacuolar proteases*	1.5c ± 0.3	1.6c ± 0.4	2.94d ± 0.8	1.5c ± 0.4	0.8b ± 0.2	n.f.a	0.6b ± 0.1	1.4c ± 0.2
*Vacuolar proteases*	0.9a ± 0.1	1.3c ± 0.2	0.9a ± 0.2	1.0ab ± 0.2	0.9a ± 0.1	2.6d ± 0.2	1.2bc ± 0.1	2.3d ± 0.2
*Nucleases*	0.18c ± 0.01	0.3de ± 0.1	0.25cd ± 0.04	0.18c ± 0.1	0.43e ± 0.02	n.f.a	0.29d ± 0.02	0.06b ± 0.02

**Table 2 microorganisms-07-00542-t002:** List of the most relevant apoptosis-related proteins identified in *S. cerevisiae* P29 and G1 under each study condition (PC: pressure condition, NPC: non-pressure condition) and sampling time (T1: middle of the second fermentation, T2: 1 month after it). Proteins showing a high content are highlighted in bold. a–g Different superscript letters indicate significant differences of protein content in each condition at 0.05 level according to Fisher’s least significant difference procedure. Proteins showing significant differences in all conditions are marked with an asterisk *. n.f.; not found. ns.; not significant.

Condition		PCT1				PCT2				NPCT1				NPCT2			
Strain		P29		G1		P29		G1		P29		G1		P29		G1	
Protein	Molecular Function	^h^ mol%	^i^ SD	mol%	SD	mol%	SD	mol%	SD	mol%	SD	mol%	SD	mol%	SD	mol%	SD
Bir1p	Subunit of chromosomal passenger complex	n.f.ns		n.f.ns		n.f.ns		n.f.ns		n.f.ns		0.005ns	0.003	n.f.ns		n.f.ns	
**Cpr3p** *	Cyclophilin C mitochondrial	0.098 ^b^	0.001	0.3 ^d^	0.1	0.0977 ^b^	0.0003	n.f. ^a^		0.081 ^ab^	0.005	0.098 ^b^	0.001	0.2 ^cd^	0.1	0.166 ^c^	0.003
Esp1p	Separin	n.f.ns		0.007ns	0.004	n.f.ns		n.f.ns		0.005ns	0.002	n.f.ns		n.f.ns		n.f.ns	
Fis1p	Mitochondria fission 1 protein	n.f. ^a^		n.f. ^a^		n.f. ^a^		n.f. ^a^		0.021 ^a^	0.007	n.f. ^a^		0.13 ^b^	0.05	n.f. ^a^	
**Mca1p** *	Metacaspase-1	n.f. ^a^		n.f. ^a^		0.11 ^c^	0.01	n.f. ^a^		0.05 ^b^	0.02	0.05 ^b^	0.02	0.137 ^d^	0.008	n.f. ^a^	
Nma111p	Pro-apoptotic serine protease	n.f.ns		0.02ns	0.01	n.f.ns		n.f.ns		0.011ns	0.006	n.f.ns		n.f.ns		n.f.ns	
**Oye2p** *	NADPH dehydrogenase 2	0.28 ^c^	0.03	0.4 ^cd^	0.1	0.4 ^d^	0.1	n.f. ^a^		0.148 ^b^	0.001	0.13 ^b^	0.07	0.131 ^b^	0.003	n.f. ^a^	
Oye3p *	NADPH dehydrogenase 3	n.f. ^a^		n.f. ^a^		n.f. ^a^		n.f. ^a^		0.04 ^b^	0.02	0.012 ^a^	0.006	0.09 ^c^	0.01	0.5 ^d^	0.1
**Por1p** *	Mitochondrial outer membrane protein porin 1	n.f. ^a^		n.f. ^a^		0.5 ^e^	0.1	n.f. ^a^		0.09 ^b^	0.05	0.05 ^ab^	0.02	0.32 ^d^	0.07	0.15 ^c^	0.05
Rny1p *	Ribonuclease T2-like	n.f. ^a^		n.f. ^a^		0.100 ^e^	0.003	n.f. ^a^		0.016 ^c^	0.004	0.011 ^b^	0.004	0.08 ^d^	0.01	n.f. ^a^	
**Tdh2p** *	Glyceraldehyde-3-phosphate dehydrogenase 2	0.6 ^b^	0.1	0.79 ^c^	0.05	0.90 ^c^	0.05	2.2 ^f^	0.1	0.36 ^a^	0.03	1.12 ^d^	0.07	1.1 ^d^	0.1	1.31 ^e^	0.04
**Tdh3p** *	Glyceraldehyde-3-phosphate dehydrogenase 3	0.72 ^b^	0.02	0.98 ^c^	0.06	1.06 ^cd^	0.01	2.27 ^g^	0.09	0.36 ^a^	0.06	1.19 ^e^	0.05	1.14 ^de^	0.04	1.4 ^f^	0.1
Ysp2p	GRAM domain-containing protein	n.f.ns		n.f.ns		n.f.ns		n.f.ns		0.003ns	0.002	n.f.ns		n.f.ns		n.f.ns	

^h^ Average values of protein content (mol %) and ^i^ standard deviation (SD).

**Table 3 microorganisms-07-00542-t003:** List of over-represented and specific proteins related to apoptosis and autolysis under CO_2_ overpressure condition (PC) detected at the middle of the second fermentation (T1) and 1 month after it (T2), in *S. cerevisiae* P29 and G1.

Strain		*S. cerevisiae* P29		*S. cerevisiae* G1	
**Sampling time**		T1	T2	T1	T2
***Apoptosis proteins***		-	Oye2p ^a^ (3.3)	Cpr3p (2.8), Esp1p (specific), Nma111p (specific), Oye2p (3.1)	
***Autolysis proteins***	*Glucanases*	-	-	Exg2p (2.4)	
	*Mannosidases*	-	Ams1p (specific)	-	
	*Vacuolar proteases*	-	-	Ape1p (2.9), Ybr139wp (2.4)	
	*Non-vacuolar proteases*	-	Pre6p (specific), Pre7p (specific), Rpn11p (specific)	Ape2p (3.3), Esp1p (specific), Lap3p (2.4), Mas2p (3.2), Nma111p (specific), Pre10p (specific), Pre4p (2.4), Ste24p (specific), Ubp6p (specific)	
	*Nucleases*	Dna2p (specific)	-	Vma1p (2.1), Ybl055cp (specific)	

^a^ Fold change of protein content (PC/NPC) and specific proteins under PC are shown in brackets.

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
