# Peer review of "First Proteomic Approach to Identify Cell Death Biomarkers in Wine Yeasts during Sparkling Wine Production"

_microorganisms, 2019, doi:10.3390/microorganisms7110542_

Round 1

Reviewer 1 Report

Review report

This work was well planned, and the analytical approach well executed. The work is interesting, and the results have direct implications in the context of wine production since some potential markers under CO2 overpressure conditions are purposed. Besides, two industrial yeast strains (P29 and G1) used commonly in post-fermentative processes were evaluated. Therefore, the manuscript entitled "Identification of potential targets to induce cell death in wine yeasts during sparkling wine production" has enough quality to be published.

Minor comments:

-    In order to avoid mistakes, different letters should be used to show statistically significant differences and food notes in Tables 2, 3, 5 and 6.

Author Response

This work was well planned, and the analytical approach well executed. The work is interesting, and the results have direct implications in the context of wine production since some potential markers under CO2 overpressure conditions are purposed. Besides, two industrial yeast strains (P29 and G1) used commonly in post-fermentative processes were evaluated. Therefore, the manuscript entitled "Identification of potential targets to induce cell death in wine yeasts during sparkling wine production" has enough quality to be published.

Minor comments:

-In order to avoid mistakes, different letters should be used to show statistically significant differences and food notes in Tables 2, 3, 5 and 6.

Thank you for the comment and suggestion, the letters of food notes were changed in the tables to avoid confusion.

Reviewer 2 Report

Dear Authors,

I have carefully read the submitted paper. The choose of the subject – dealing with apoptosis, autolysis, proteome, yeast, sparkling wine, CO2 overpressure- seems to be justified, as few works have addressed the point so far in literature. The experimental design is clearly presented, and results are generally in line. However, some important changes in data analysis and presentation should be done to make the article suitable for publication.

Some general remarks:

Overall, one important point is that the development of second fermentation by the two studied yeasts is not sufficiently described: the duration of the process (which is not stated) may influence yeast behavior, moreover the final wine should be characterized at least in terms of residual sugars (if any), volatile acidity, pH, malic and lactic acid (reporting these data would clarify variability between the 3 bottles).

Data reported in Table 1 lack of statistical analysis. This should be added (e.g. ANOVA considering strain factor on 2 levels, pressure factor on 2 levels, time of sampling factor on 2 levels). This would also simplify data presentation.

As another general remark, the data presentation is not easy to follow because of the many complex tables. I would suggest transforming some tables into graphs or, either way, to highlight (e.g. bold) in tables 2 and 3 relevant data that are also discussed in the text. The same for table 6 and 7. These tables should also include a column reporting GO annotation of the proteins, to make them more informative.

Moreover, a table equivalent to table 5 is missing for autolysis proteins, this should be added, or the point should be justified.

Furthermore, figure 3 is reported but never ever commented. Either the Authors add a consistent commentary on the interaction network and its significance, or the figure should be removed. Please improve this part.

Concerning figure 4 and its comment, few words should be added about strain effect: Authors claim that the work aims at identifying markers of apoptosis, but if the same proteins show opposite correlation with cell viability depending on the strain, their interest is only related to these specific strains. A proof of concept with other stains and other conditions would be needed to clarify these hypotheses.

Concerning the title, what do the authors mean with “potential targets to induce cell death in wine yeasts during sparkling wine production“ …? How would this specific induction be done? How many markers (non strain-dependent) do they identify at the end of the work? How do they propose to induce cell death? Please clarify the point or amend the title.

Some more specific remarks:

Par. 2.12.2: please specify inoculation protocol (active dry yeast? Pre-cultures?)

Par 2.4: to help the reader, some main information of the proteomic experiment should be briefly outlined, then referring for details to the cited publications.

Lines 125-126: the first sentence, referring to protein analysis, is misleading at the beginning of a paragraph discussing cell viability data. I would suggest adding some secondary headlines within “results and discussion”, to separate comments about cell viability form those about protein content.

Figure 1 should be amended to make clear the data of PCT2: either using a log scale (best when reporting cell count data) or, in case, two different scales for T1 and T2.

Line 142: a general introduction about the number of proteins detected, the number of proteins related to apoptosis and autolysis in SGD etc. should be added.

Conclusions: line 58 – what do the Authors mean with “targets in order to accelerate cell death”?

Author Response

Dear Authors,

I have carefully read the submitted paper. The choose of the subject – dealing with apoptosis, autolysis, proteome, yeast, sparkling wine, CO2 overpressure- seems to be justified, as few works have addressed the point so far in literature. The experimental design is clearly presented, and results are generally in line. However, some important changes in data analysis and presentation should be done to make the article suitable for publication.

Some general remarks:

Overall, one important point is that the development of second fermentation by the two studied yeasts is not sufficiently described: the duration of the process (which is not stated) may influence yeast behavior, moreover the final wine should be characterized at least in terms of residual sugars (if any), volatile acidity, pH, malic and lactic acid (reporting these data would clarify variability between the 3 bottles).

-The second fermentation kinetics performed by both strains has been added to results and discussion section (now Figure 1).

-Final wine composition:

Wine P29: 11.6 % v/v ethanol, 0.3 g/L of reducing sugars, 0.2 g/L of volatile acidity, 5.3 g/L of total acidity, 3.3 pH, 1.89 g/L malic acid, 0.1 g/L lactic acid.

Wine G1: 11.4 % v/v ethanol, 0.3 g/L of reducing sugars, 0.3 g/L of volatile acidity, 4.3 g/L of total acidity, 3.4 pH, 0.2 g/L malic acid, 1.2 g/L lactic acid.

This wine composition has been included in results and discussion section (lines 145-149), and in addition, a section into materials and methods about the quantification of these general wine parameters was added (lines 106-110).

Data reported in Table 1 lack of statistical analysis. This should be added (e.g. ANOVA considering strain factor on 2 levels, pressure factor on 2 levels, time of sampling factor on 2 levels). This would also simplify data presentation.

Statistical analysis has been added to the Table 1.

Moreover, due to the complexity of the table, frequency values were removed and the most relevant data were added to the manuscript (lines 179-185) in the section “study of the proteome”, in order to simplify the data presentation.

As another general remark, the data presentation is not easy to follow because of the many complex tables. I would suggest transforming some tables into graphs or, either way, to highlight (e.g. bold) in tables 2 and 3 relevant data that are also discussed in the text. The same for table 6 and 7. These tables should also include a column reporting GO annotation of the proteins, to make them more informative.

-Tables 2 and 3, and also 5 and 6, were joined in order to simplify and decrease the number of tables. New Tables 2 and S2 show only protein content values of apoptosis and autolysis proteins, respectively, in both yeast strains. In addition, a column reporting the molecular function of each protein has been added.

Score and peptides values were included in Table S1 as supplementary material.

-In each table, the most relevant proteins were highlighted in bold and those significantly different were marked with an asterisk.

Moreover, a table equivalent to table 5 is missing for autolysis proteins, this should be added, or the point should be justified.

The autolysis proteins are shown in Table 3, along with the over-represented proteins detected for apoptosis.

For a better understanding, processes in the table are marked in bold as well as the border that divide them.

Furthermore, figure 3 is reported but never ever commented. Either the Authors add a consistent commentary on the interaction network and its significance, or the figure should be removed. Please improve this part.

The paragraph containing the comments about the interaction maps (Fig. 3 and 4) was improved and the most relevant results extracted from the interaction maps were highlighted (lines 219-239). In addition, new Figure 4 was added to the discussion section (lines 373-375).

Concerning figure 4 and its comment, few words should be added about strain effect: Authors claim that the work aims at identifying markers of apoptosis, but if the same proteins show opposite correlation with cell viability depending on the strain, their interest is only related to these specific strains. A proof of concept with other strains and other conditions would be needed to clarify these hypotheses.

The use of these yeast strains represents a first approach to understand their behavior under these conditions and identify possible biomarkers in order to improve the industrial process. Experiments with other yeast strains and conditions will be carried out in future works.

Concerning the title, what do the authors mean with “potential targets to induce cell death in wine yeasts during sparkling wine production“ …? How would this specific induction be done? How many markers (non strain-dependent) do they identify at the end of the work? How do they propose to induce cell death? Please clarify the point or amend the title.

-What we tried to say with this sentence was that the genes encoding proteins found highly represented under studied conditions, can be used to promote cell death and further autolysis. This study provides a background knowledge about proteomic profile that could be used as a criteria for the yeast strains selection for sparkling wine elaboration.

-This induction could be carried out with a yeast pre-adaptation to medium conditions. However, this study represent a first approach to understand the proteomic response of these strains to these conditions. The possible induction or cell death promotion would be performed in future works once the corresponding genes are studied, as well as their possible use in the industry.

The use of genetic engineering of wine yeasts in order to accelerate aging via inducing autolysis has been proposed (“Induction of autophagy by second-fermentation yeasts during elaboration of sparkling wines” by Cebollero and Gonzalez, 2006 https://aem.asm.org/content/72/6/4121). Moreover, the use of mutants have been reported by numerous authors (“Yeast autolytic mutants potentially useful for sparkling wine production” by Gonzalez et al., 2003 https://www.sciencedirect.com/science/article/pii/S0168160502003896?via%3Dihub; “Evidence for yeast autophagy during simulation of sparkling wine aging: a reappraisal of the mechanism of yeast autolysis in wine” by Cebollero et al., 2005 https://aiche.onlinelibrary.wiley.com/doi/full/10.1021/bp049708y; “Effect of accelerated autolysis of yeast on the composition and foaming properties of sparkling wines elaborated by a champenoise method” by Nunez et al., 2005 https://pubs.acs.org/doi/10.1021/jf050191v; “Deletion of BCY1 from the Saccharomyces cerevisiae Genome Is Semidominant and Induces Autolytic Phenotypes Suitable for Improvement of Sparkling Wines” by Tabera et al., 2006 https://aem.asm.org/content/72/4/2351).

-Biomarkers identified under pressure non strain-dependent: Oye2p, Bgl2p, Exg1p and Pep4p.

For a better understanding and avoid confusion, the title was modified.

Some more specific remarks:

Par. 2.1-2.2: please specify inoculation protocol (active dry yeast? Pre-cultures?)

-Prior to yeast acclimation in the pasteurized must, yeast cells of each strain was growth in YPD medium at 21 ℃ and 48 h. This information was added to the materials and methods section (line 81).

-Active dry yeasts were not used.

Par 2.4: to help the reader, some main information of the proteomic experiment should be briefly outlined, then referring for details to the cited publications.

Main information about proteomic analysis was added to materials and methods section (lines 113-118).

Lines 125-126: the first sentence, referring to protein analysis, is misleading at the beginning of a paragraph discussing cell viability data. I would suggest adding some secondary headlines within “results and discussion”, to separate comments about cell viability form those about protein content.

Headlines were provided to separate cell viability from the study of the proteome. Also the first sentence that could lead to confusion was removed.

Figure 1 should be amended to make clear the data of PCT2: either using a log scale (best when reporting cell count data) or, in case, two different scales for T1 and T2.

This figure has been modified using two different scales to make clear the data of PCT2.

Line 142: a general introduction about the number of proteins detected, the number of proteins related to apoptosis and autolysis in SGD etc. should be added.

Conclusions: line 58 – what do the Authors mean with “targets in order to accelerate cell death”?

-A general introduction reporting the total number of apoptosis and autolysis proteins found in this work as well as the total number detected in SGD has been added at the beginning of the section 3.2 in results and discussion (lines 173-178).

-The sentence means proteins, mainly those detected highly represented under pressure or specific of this condition, whose genes could be interesting for future works to promote cell death and further autolysis in wine yeasts.

Reviewer 3 Report

This study reports the apoptosis and autolysis-related proteome of two industrial yeast strains (P29 CECT 11770) isolated from INCAVI (Catalan Institute of Vines and Wines); and a flor yeast strain (G1 ATCC MYA-2451), isolated from a wine velum from PDO Montilla-Moriles (Spain), used in sparkling wines fermentation process. The authors compared proteome in order to detect potential markers under CO2 overpressure conditions (typical of sparkling wine elaboration process) as well as to improve and accelerate this industrial process.

The main objective of the paper is interesting, and it shows a lot of results. By the way it appears boring and with a poor discussion.

To ease readers’ comprehension, table 5 and table 6 should be presented as supplementary data. Moreover, a statistical analysis of proteins should be reported only for the proteins useful to differentiate the strains and the process.

To improve the discussion some papers should be considered:

Tofalo, G. Perpetuini, P. Di Gianvito, G. Arfelli, M. Schirone, A. Corsetti, G. Suzzi (2016).

Characterization of specialized flocculent yeasts to improve sparkling wine fermentation. Journal of Applied Microbiology 120, 1574—1584

Perpetuini, P. Di Gianvito, G. Arfelli, M. Schirone, A. Corsetti, R. Tofalo, G. Suzzi (2016). Biodiversity of autolytic ability in flocculent Saccharomyces cerevisiae strains suitable for traditional sparkling wine fermentation. Yeast 33, 03–312 Garofalo, C. Berbegal, F. Grieco, M. Tufariello, G. Spano, V. Capozzi (2018) Selection of indigenous yeast strains for the production of sparkling wines from native Apulian grape varieties. International Journal of Food Microbiology. 20; 7-17

Please shorten materials and methods considering already published methods.

Please fix table 1: parenthesis should be closed in the same line as they open.

Paper should be accepted with major revisions.

Author Response

This study reports the apoptosis and autolysis-related proteome of two industrial yeast strains (P29 CECT 11770) isolated from INCAVI (Catalan Institute of Vines and Wines); and a flor yeast strain (G1 ATCC MYA-2451), isolated from a wine velum from PDO Montilla-Moriles (Spain), used in sparkling wines fermentation process. The authors compared proteome in order to detect potential markers under CO2 overpressure conditions (typical of sparkling wine elaboration process) as well as to improve and accelerate this industrial process.

The main objective of the paper is interesting, and it shows a lot of results. By the way it appears boring and with a poor discussion.

To ease readers’ comprehension, table 5 and table 6 should be presented as supplementary data. Moreover, a statistical analysis of proteins should be reported only for the proteins useful to differentiate the strains and the process.

-In order to simplify the results, tables 5 and 6 were joined. The new table S2 shows the total list autolysis proteins identified in both strains as supplementary data. The most relevant proteins in terms of content and statistically significant are highlighted.

-In addition, Principal Component Analysis based on those proteins from apoptosis and autolysis significantly different was performed in Figure 6.

To improve the discussion some papers should be considered:

Tofalo, G. Perpetuini, P. Di Gianvito, G. Arfelli, M. Schirone, A. Corsetti, G. Suzzi (2016).

Characterization of specialized flocculent yeasts to improve sparkling wine fermentation. Journal of Applied Microbiology 120, 1574—1584

Perpetuini, P. Di Gianvito, G. Arfelli, M. Schirone, A. Corsetti, R. Tofalo, G. Suzzi (2016). Biodiversity of autolytic ability in flocculent Saccharomyces cerevisiae strains suitable for traditional sparkling wine fermentation. Yeast 33, 03–312

Garofalo, C. Berbegal, F. Grieco, M. Tufariello, G. Spano, V. Capozzi (2018) Selection of indigenous yeast strains for the production of sparkling wines from native Apulian grape varieties. International Journal of Food Microbiology. 20; 7-17

The references were added to the discussion (line 414).

Please shorten materials and methods considering already published methods.

According to other reviewers’ comments, some main information about proteomic analysis was added (lines 113-118) and referring for details to the cited publications. However, it was not possible to shorten more this section since it includes methods not reported so far.

Please fix table 1: parenthesis should be closed in the same line as they open.

The parenthesis were closed in the same line.

Paper should be accepted with major revisions.

Reviewer 4 Report

Identification of potential targets to induce cell death  in wine yeasts during sparkling wine production

 In the present manuscript the authors have used two industrial yeast strains, one from sparkling wine production and the other from biological aging of wine, to conduct secondary fermentations of sparkling wine. The fermentations were performed under typical condition (under CO2 pressure, that results from the fermentation) and without CO2 pressure. They then analyzed cell viability, at two time points, and perform a proteomic analysis, at the same time points, looking at the proteins annotated as involved in apoptosis or autolysis.

Although the authors explain that they think the faster autolysis of flor yeast could be advantageous to accelerate sparkling wine production, they make no other consideration on the adequacy of using flor yeast to produce sparkling wine. Would this be a realistic alternative? Did the flour yeast  led to an acceptable quality sparkling wine?

On the other hand,  the other output from the work according to the authors, the possibility of producing genetically modified yeasts with faster cell death and autolysis for wine production, is currently not allowed in Europe and generally not accepted by the consumers. So, I think that the option to use this strain in the current work should be more sustained.

Another issue of using two strains from different application/environments for conducting the same process, is that the yeast which is not usually employed in this process,  the flor yeast, is more well adapted to very different conditions, namely the presence of O2 leading to the prevalence of an oxidative metabolism, in contrast with the fermentation metabolism and high CO2 concentrations in sparkling wine production. This differences in environmental conditions, may lead to a quite different response that may not only reflect differences in the resistance of the strain to cell death and autolysis, but can also lead to different metabolic adaptation that can confuse the results. This is especially relevant as some of the proteins involved in cell death are also involved in metabolic pathways, and this should be taken in consideration when the comparisons between the two strains are made.

Also, due to considerable differences in resistance, at the time points of analysis, the two yeasts populations have very different viabilities, so  the comparisons are made between cells in very different physiological conditions which may also be responsible for some of the differences observed. That is, from this work it is not clear if the differences observed between the two strains, are due to strain-specific responses, or to the fact that the samples were taken at different stages of the cell death/autolysis process. This should be taken in consideration in interpretation and discussion of results.

Also, to it is not very clear what useful information can result from the comparison between the PC and NPC conditions using two strains with very different responses, the discussion resulting a little descriptive and not allowing for any solid conclusions – in fact there are so many variables between samples that is difficult to conclude much. It would be important to have more time-points to understand what is happening in the fermentation to better understand the dynamic of the process, even if a proteomic analysis would not be done in the other time points. Also, since no complementary experiments are performed to validate some of the conclusions made much of the discussion remains speculative.

The manuscript has some interesting information and the methods seem adequate but treatment/discussion of results should be more systematized and care should be taken when performing comparisons, so they can be informative.

Specific comments:

Pag 1, lines 22-23 -  “Results suggest an important effect of CO2 overpressure on cell viability” – this is not true for the P29 strain at T1, which seems to perform better under PC.

 Pag 1, lines 39-41 -  “Among the types of PCD, apoptosis appears to be a programmed cell death (PCD) subroutine characterized by specific morphologic and biochemical features, and different pro-apoptotic factors [3].” this is not right, apoptosis can occur as a consequence of environmental stress and not in a physiological scenario as is the case of PCD. Here, I would say that “PCD can occur with apoptotic features”. In its current form this sentence is misleading suggesting that apoptosis is only occurring in the context of PCD, when in fact it is a form or RCD that can be totally independent of a PCD process.

Lines 44-45 – “The discovery of apoptosis in yeasts has suggested that other forms of PCD” - PCD should be replaced by RCD. Please see the paper from Lorenzo Galluzzi et al in Cell Death & Differentiation (2018), for a up to date use of nomenclature. PCD is a particular  form of RCD that occurs only in physiological contexts

Lines 125-126 – “Proteins involved in apoptosis and autolysis processes were identified in both yeast strains under each study condition (PC and NPC) and sampling time (T1 and T2).” – in MM authors say T1 sampling time is when the pressure reached 3 bars - it would be important to know how many days took to reach this point for each strain. Was it reached at the same day of secondary fermentation for both strains? This information is important to understand also the dynamic of the fermentation.

Lines 128-129 – “Focusing on each strain separately, remarkable differences can be noticed in samples subjected to CO2 pressure (PCT1 and PCT2), where the yeast strain P29 outperformed G1, reaching more than double in both sampling times.” –  P29 also outperformed G1 in NPC conditions at T1, which shows it is probably more adapted to the medium not only the pressure. In fact, regarding the pressure it  seems it performs even better under PC.

Line 147 – “affects to flor yeast” change to “affects flor yeast”

Pag 5, line 18 – “GAPDH (glyceraldehyde-3-phosphate dehydrogenase) isoenzymes Tdh2p and Tdh3p highlighted” – these enzymes are crucial enzymes of glycolysis and their content can also modulate glycolytic flux. Since we have no information regarding other glycolytic enzymes it is difficult to conclude if this increase can be related to cell death or to metabolism.

Pag 12, line 35-37 – “This vacuolar protease has been demonstrated to be the main enzyme involved in yeast autolysis and responsible for 80% of the nitrogen release [54], although there is no correlation among proteinase A, cell death and autolysis” – in fact, the role of Pep4 in cell death has been addressed by several groups, and it has been shown to play a protective role under acetic acid induced apoptosis (Pereira C, Chaves S, Alves S, Salin B, Camougrand N, Manon S, Sousa MJ, Côrte-Real M. Mitochondrial degradation in acetic acid-induced yeast apoptosis: the role of Pep4 and the ADP/ATP carrier. Mol Microbiol 76(6):1398-410) and  during chronological aging,  where deletion of PEP4 resulted in both apoptotic and necrotic cell death (Carmona-Gutiérrez D, Bauer MA, Ring J, Knauer H, Eisenberg T, Büttner S, Ruckenstuhl C, Reisenbichler A, Magnes C, Rechberger GN, Birner-Gruenberger R, Jungwirth H, Fröhlich KU, Sinner F, Kroemer G, Madeo F. The propeptide of yeast cathepsin D inhibits programmed necrosis. Cell Death Dis. 2011 May 19;2:e161. doi: 10.1038/cddis.2011.43.)

Figure 4 – decrease the size of the squares and increase the text size  in the figure.

Author Response

In the present manuscript the authors have used two industrial yeast strains, one from sparkling wine production and the other from biological aging of wine, to conduct secondary fermentations of sparkling wine. The fermentations were performed under typical condition (under CO2 pressure, that results from the fermentation) and without CO2 pressure. They then analyzed cell viability, at two time points, and perform a proteomic analysis, at the same time points, looking at the proteins annotated as involved in apoptosis or autolysis.

Although the authors explain that they think the faster autolysis of flor yeast could be advantageous to accelerate sparkling wine production, they make no other consideration on the adequacy of using flor yeast to produce sparkling wine. Would this be a realistic alternative? Did the flour yeast led to an acceptable quality sparkling wine?

-The use of flor yeast would represent a real alternative, not only considering the results obtained in this study, but also other which report the impact of this yeast strain on sparkling wine quality (“Use of a flor yeast strain for the second fermentation of sparkling wines: effect of endogenous CO2 over-pressure on the volatilome” by Martínez-García et al., 2019, in press https://doi.org/10.1016/j.foodchem.2019.125555). This reference has been added to the discussion section (lines 414-416).

Furthermore, this yeast strain is characterized by both its ethanol resistance during fermentation, since it was isolated from a wine with 14.5 % v/v and flocculation capacity. Additionally, it has not been reported to negatively affect the aroma of the wine.

All in all, flor yeasts would be result a good candidate for the innovation and diversification in sparkling wine-making.

On the other hand,  the other output from the work according to the authors, the possibility of producing genetically modified yeasts with faster cell death and autolysis for wine production, is currently not allowed in Europe and generally not accepted by the consumers. So, I think that the option to use this strain in the current work should be more sustained.

-This work represents a first approach to identify potential cell death biomarkers under studied conditions. The genetic study of this research has been planned for future works in order to see if these genes affect positively to wine quality and they can promote and accelerate the acquisition of the organoleptic properties during aging period. After this genetic study, the selection of yeast strains with this corresponding proteomic profile could be used by winemaking, considering the current regulations regarding the use of modified strains.

-Although the production of genetically modified yeasts is not allowed in Europe, the wide knowledge obtained from these studies would be useful for winemakers since it would give them an insight about the yeast behaviour or provide them with a proteomic profile under these special conditions. This profile can be used to select suitable yeast strains for sparkling wine elaboration.

-Moreover, it should be mentioned that flor yeasts are also fermentative and they have not been reported to produce strange aroma.

Another issue of using two strains from different application/environments for conducting the same process, is that the yeast which is not usually employed in this process, the flor yeast, is more well adapted to very different conditions, namely the presence of O2 leading to the prevalence of an oxidative metabolism, in contrast with the fermentation metabolism and high CO2 concentrations in sparkling wine production. This differences in environmental conditions, may lead to a quite different response that may not only reflect differences in the resistance of the strain to cell death and autolysis, but can also lead to different metabolic adaptation that can confuse the results. This is especially relevant as some of the proteins involved in cell death are also involved in metabolic pathways, and this should be taken in consideration when the comparisons between the two strains are made.

-We agree with your comment. The fact that some proteins are also involved in metabolism such as the GAPDH proteins has already been taken into account in the discussion (lines 295-298).

Also, due to considerable differences in resistance, at the time points of analysis, the two yeasts populations have very different viabilities, so the comparisons are made between cells in very different physiological conditions which may also be responsible for some of the differences observed. That is, from this work it is not clear if the differences observed between the two strains, are due to strain-specific responses, or to the fact that the samples were taken at different stages of the cell death/autolysis process. This should be taken in consideration in interpretation and discussion of results.

-We agree with your comment, since these two parameters could affect the results.

It should be mentioned that the aim of this study is to analyse the proteomic profile along the second fermentation for the first time in a typical sparkling wine strain used as a control (S. cerevisiae P29). On the other hand, we also compared this profile with a strain not adapted to these conditions as a flor yeast (G1), since we think that due to the advantages mentioned before (ethanol resistance, flocculation ability and contribution to wine aroma), it could represent a real alternative in this field, leading to an improvement and elaboration of sparkling wines.

-The differences between the strains could be due to the different response of each strain to study conditions, since one strain is more adapted to this environment than the other. Apart from this, the sampling points can also affect since the environmental conditions are different in terms of pressure, ethanol content or nutrients availability. These issues has been taken into account in the discussion (lines 157-164) and moreover, the PCA provided in Fig. 6 showed relevant information about these differences.

Also, to it is not very clear what useful information can result from the comparison between the PC and NPC conditions using two strains with very different responses, the discussion resulting a little descriptive and not allowing for any solid conclusions – in fact there are so many variables between samples that is difficult to conclude much. It would be important to have more time-points to understand what is happening in the fermentation to better understand the dynamic of the process, even if a proteomic analysis would not be done in the other time points. Also, since no complementary experiments are performed to validate some of the conclusions made much of the discussion remains speculative.

-This article represent a first approach to identify possible biomarkers under pressure. Our group and the INCAVI colleagues decided that these points are enough for a first analysis. Furthermore, it should be mentioned that this type of study has not been reported so far. The same approach has already been used for publishing other articles where after investigating the flor yeast proteome and identifying interesting proteins (“Stress responsive proteins of a flor yeast strain during the early stages of biofilm formation” by Moreno-García et al., 2016 https://www.sciencedirect.com/science/article/pii/S135951131630023X?via%3Dihub), these were used to evaluate their contribution to the biofilm ability through the mutant production (“Study of the role of the covalently linked cell wall protein (Ccw14p) and yeast glycoprotein (Ygp1p) within biofilm formation in a flor yeast strain” by Moreno-García et al., 2018 https://doi.org/10.1093/femsyr/foy005).

Additional experiments and more points along the second fermentation are being studied for future works.

-For a better understanding and according to reviewers’ comments, the title of the article has been modified.

The manuscript has some interesting information and the methods seem adequate but treatment/discussion of results should be more systematized and care should be taken when performing comparisons, so they can be informative.

- Thank you for your comment, the discussion has been carefully revised taking into account all the suggestions, which have considerably improved the quality work.

Specific comments:

Pag 1, lines 22-23 - “Results suggest an important effect of CO2 overpressure on cell viability” – this is not true for the P29 strain at T1, which seems to perform better under PC.

The sentence was modified and clarified in the abstract.

Pag 1, lines 39-41 -  “Among the types of PCD, apoptosis appears to be a programmed cell death (PCD) subroutine characterized by specific morphologic and biochemical features, and different pro-apoptotic factors [3].” this is not right, apoptosis can occur as a consequence of environmental stress and not in a physiological scenario as is the case of PCD. Here, I would say that “PCD can occur with apoptotic features”. In its current form this sentence is misleading suggesting that apoptosis is only occurring in the context of PCD, when in fact it is a form or RCD that can be totally independent of a PCD process.

The sentence was revised and modified in order to avoid confusion.

Lines 44-45 – “The discovery of apoptosis in yeasts has suggested that other forms of PCD” - PCD should be replaced by RCD. Please see the paper from Lorenzo Galluzzi et al in Cell Death & Differentiation (2018), for a up to date use of nomenclature. PCD is a particular form of RCD that occurs only in physiological contexts.

The sentence has been modified, changing PCD by RCD. In addition, the reference Galluzzi et al. 2018 was added in the introduction (lines 43 and 47).

Lines 125-126 – “Proteins involved in apoptosis and autolysis processes were identified in both yeast strains under each study condition (PC and NPC) and sampling time (T1 and T2).” – in MM authors say T1 sampling time is when the pressure reached 3 bars - it would be important to know how many days took to reach this point for each strain. Was it reached at the same day of secondary fermentation for both strains? This information is important to understand also the dynamic of the fermentation.

The second fermentation kinetics performed by both strains has been added to results and discussion section (now Figure 1). The yeast strain P29 reached 3.3 bar at 8 days, whereas flor yeast G1 took 10 days (lines 140-144).

Lines 128-129 – “Focusing on each strain separately, remarkable differences can be noticed in samples subjected to CO2 pressure (PCT1 and PCT2), where the yeast strain P29 outperformed G1, reaching more than double in both sampling times.” –  P29 also outperformed G1 in NPC conditions at T1, which shows it is probably more adapted to the medium not only the pressure. In fact, regarding the pressure it seems it performs even better under PC.

The sentence was modified and clarified for a better understanding (lines 153-156).

Line 147 – “affects to flor yeast” change to “affects flor yeast”

The sentence was changed.

Pag 5, line 18 – “GAPDH (glyceraldehyde-3-phosphate dehydrogenase) isoenzymes Tdh2p and Tdh3p highlighted” – these enzymes are crucial enzymes of glycolysis and their content can also modulate glycolytic flux. Since we have no information regarding other glycolytic enzymes it is difficult to conclude if this increase can be related to cell death or to metabolism.

The paragraph containing the discussion of these proteins was modified and clarified. Moreover, references have been included to explain their role in metabolism (lines 294-298).

Pag 12, line 35-37 – “This vacuolar protease has been demonstrated to be the main enzyme involved in yeast autolysis and responsible for 80% of the nitrogen release [54], although there is no correlation among proteinase A, cell death and autolysis” – in fact, the role of Pep4 in cell death has been addressed by several groups, and it has been shown to play a protective role under acetic acid induced apoptosis (Pereira C, Chaves S, Alves S, Salin B, Camougrand N, Manon S, Sousa MJ, Côrte-Real M. Mitochondrial degradation in acetic acid-induced yeast apoptosis: the role of Pep4 and the ADP/ATP carrier. Mol Microbiol 76(6):1398-410) and  during chronological aging,  where deletion of PEP4 resulted in both apoptotic and necrotic cell death (Carmona-Gutiérrez D, Bauer MA, Ring J, Knauer H, Eisenberg T, Büttner S, Ruckenstuhl C, Reisenbichler A, Magnes C, Rechberger GN, Birner-Gruenberger R, Jungwirth H, Fröhlich KU, Sinner F, Kroemer G, Madeo F. The propeptide of yeast cathepsin D inhibits programmed necrosis. Cell Death Dis. 2011 May 19;2:e161. doi: 10.1038/cddis.2011.43.)

The sentence was modified and the references were added (lines 394-397).

Figure 4 – decrease the size of the squares and increase the text size in the figure.

The size of the squares was decreased and the text was increased to improve the visualization.

Round 2

Reviewer 2 Report

Dear Authors,

thanks to the changes done to respond to all the Reviewers, the work is substantially improved and now deserves publication.

Reviewer 3 Report

The paper has been improved.

Reviewer 4 Report

The authors have addressed all the comments carefully and adequality made the corresponding changes.

There are, however, still some language/typos issues that need correction.

Specific comments:

Pag 1, lines 17-18-  “Therefore, the identification of potential cell death biomarkers can contribute to the knowledge of creating a long-term strategy in order to improve and accelerate the winemaking process” –  I would suggest to simplify it to “Therefore, the identification of potential cell death biomarkers can contribute to the creation of a long-term strategy in order to improve and accelerate the winemaking process”

 Pag 1, line 22 -  “Pressure affected negatively on viability for flor yeast,”  should be better as “Pressure negatively affected viability of flor yeast” or “Pressure had a negatively effect on viability for flor yeast”

Line 81 – “cytochrome c release” should be “cytochrome c release” the c is in italics

Lines 81-82 – “The discovery of apoptosis in yeasts has suggested that other forms of RCD might regulate cell death” would be better as ““The discovery of apoptosis in yeasts has suggested that other forms of RCD might occur

Lines 96 – “This yeasts tend” change to “These yeasts tend”

Line 851 – “fermentation in two industrial.” change to “fermentation in two industrial strains.”